# Vascular Ring Surgical Repair: Re-Implantation of the Left Subclavian Artery into the Left Carotid Artery in a Pediatric Patient

**DOI:** 10.3390/diagnostics14161736

**Published:** 2024-08-10

**Authors:** María Alejandra Rodríguez Brilla, Kevin Daniel Kausen, Aboozar T. Ali, Govinda Paudel, Victor Bautista-Hernandez

**Affiliations:** 1Division of Cardio-Thoracic Surgery, Michael E. DeBakey Department of Surgery, Baylor College of Medicine at Christus Children’s Hospital of San Antonio, San Antonio, TX 78207, USA; ma.rodriguezb1@uniandes.edu.co (M.A.R.B.); kausen@student.uiwtx.edu (K.D.K.); aboozarali001@gmail.com (A.T.A.); 2School of Osteopathic Medicine, University of the Incarnate Word, San Antonio, TX 78235, USA; 3Department of Pediatric Cardiology, Baylor College of Medicine at Christus Children’s Hospital of San Antonio, San Antonio, TX 78207, USA; govinda.paudel@christushealth.org

**Keywords:** vascular ring repair, Kommerell’s diverticulum, EndoFLIP^TM^ technology, aberrant subclavian artery re-implantation, congenital heart disease

## Abstract

Vascular rings are a rare congenital anomaly of the aortic arch, in which a ring-shaped structure forms, surrounding the trachea and/or esophagus, potentially causing compression. We describe the case of a 14-month-old female patient with failure to thrive secondary to dysphagia, and a vascular ring formed by a right aortic arch, an aberrant left subclavian artery, and a left ligamentum arteriosum. Surgical repair involved ligamentum arteriosum division, Kommerell’s diverticulum obliteration, and left subclavian artery re-implantation into the left carotid artery. Endoscopy and EndoFLIP^TM^ evaluated the intraoperative improvement in esophageal narrowing and impedance, respectively. The postoperative period was uneventful, and follow-up visits demonstrated dysphagia resolution and a patent re-implanted left subclavian artery.

Vascular rings have an estimated prevalence of 1 in 10,000 live births [1]. These rare congenital anomalies of the aortic arch encircle and compress the trachea and/or esophagus, presenting as nonspecific symptoms that vary in severity depending on the affected structure and the degree of compression [1,2,3]. Early surgical intervention is crucial in symptomatic vascular rings [1]. Surgical repair focuses on relieving tracheal and/or esophageal compression by dividing the vascular ring, thereby providing symptomatic relief and preventing serious complications, such as sudden death or residual tracheobronchial damage [1,3]. However, determining symptoms in infants and pediatric patients can be challenging due to symptom overlap with common conditions in these age groups. Herein, we present a case of surgical repair of a vascular ring in which an intraoperative evaluation of esophageal compression release was performed with endoscopy and EndoFLIP^TM^ (Medtronic, Minneapolis, MN, USA) technology.

A 5-month-old female patient was being evaluated for failure to thrive secondary to dysphagia while on a liquid diet, requiring feeding aids (nasogastric tube). Past medical history accounts for in utero umbilical hemorrhage at 34 weeks’ gestation and intrauterine growth restriction, without any cardiovascular anomalies identified in fetal morphology scans. A barium swallow test was performed, which revealed a posterior indentation of the proximal thoracic esophagus (Figure 1).

This finding prompted a Computed Tomography Angiography (CTA), which revealed a vascular ring formed by a right aortic arch (RAA), an aberrant left subclavian artery (LSA) originating from an aneurysmal dilation at its base, the so-called Kommerell’s diverticulum (KD), and a left ligamentum arteriosum (LLA) (Figure 2) (Appendix A).

The patient was then placed on a gastrostomy tube and followed up until 14 months of age, showing persistent dysphagia and weighing 8.2 kg. Hence, the decision was made to perform a surgical repair of the vascular ring. Prior to the surgical incision, an endoscopic and EndoFLIP^TM^ evaluation were conducted, demonstrating an esophageal narrowing at the level of the vascular ring and an abnormal increase in esophageal impedance, respectively. A limited left posterior thoracotomy was performed, allowing the visualization of a left-sided esophagus with a RAA originating from behind, and an aberrant LSA exiting from a KD. The LLA, which was observed compressing the esophagus, was divided. The KD was obliterated, with two purse-string sutures, and the aberrant LSA was dissected and divided from the aorta, then re-implanted into the left carotid artery (LCA) via an end-to-side anastomosis. Following the closure of the chest, another endoscopic evaluation and EndoFLIP^TM^ assessment were performed, which evidenced an improvement in esophageal narrowing and impedance, respectively. The patient’s recovery was uneventful, and she was discharged on postoperative day 4. Follow-up visits demonstrated symptomatic improvement without postoperative complications, and she was meeting her growth milestones. A recent CTA performed 2 years post-procedure revealed a patent re-implanted LSA into the LCA and no tracheal deformities (Figure 3) (Appendix A).

We describe the case of a 14-month-old female patient with a vascular ring who underwent successful surgical repair. The intraoperative demonstration of esophageal compression relief was achieved using endoscopic and EndoFLIP™ technology. The re-implantation of the LSA into the LCA was performed, and the patency of the re-implanted artery, without evident tracheal deformities or esophageal narrowing, was confirmed by the two-year postoperative CTA.

Vascular rings are congenital aortic arch anomalies that result in the compression of the trachea, esophagus, or both [2]. The most reported types of vascular rings are the double aortic arch (DAA) and the RAA with aberrant LSA and LLA, the latter being usually associated with a Kommerell’s diverticulum [4]. Patients presenting with a DAA commonly exhibit symptoms in the first year of life due to severe respiratory compromise secondary to significant tracheal compression [4]. However, patients with an RAA with an aberrant LSA and LLA typically start having symptoms within the first few years of life, usually presenting as a combination of respiratory and esophageal manifestations [4]. Hence, symptom presentation depends on the type of vascular ring, the degree of compression, and the patient’s age, ranging from respiratory distress in newborns to feeding difficulties in older children [2,5]. Evidencing symptoms in small infants can pose a significant diagnostic challenge due to the overlap with common manifestations observed in this age group, such as stridor, wheezing, recurrent respiratory infections, vomiting, reflux, and dysphagia [1,2,3,6,7].

Traditionally, the initial evaluation of vascular rings involves chest X-ray, barium swallow, and echocardiography [1,2,3,4]. However, cross-sectional imaging techniques such as Multidetector Computed Tomography (MDCT) and Magnetic Resonance Imaging (MRI) are necessary for detailed anatomical assessment. MDCT offers excellent visualization of vascular anatomy and is often preferred over MRI due to its speed and clarity without requiring sedation [1]. Although the prenatal diagnosis of vascular rings has improved with fetal echocardiography and advanced 3D sonography, which provide clearer views of the aortic arch position relative to the trachea, the rate of missed diagnosis and misdiagnosis is still very high [1,8]. A recent study by Zhou et al. (2023) demonstrated that dynamic sequential cross-sectional observation (SCS) can accurately diagnose congenital vascular rings in the prenatal period, although it supports expectant management until birth [4,7].

Early surgical repair of symptomatic vascular rings is crucial, aiming to relieve the compression while reducing the associated potential complications [1]. Yoshimura et al. (2020) discuss the various surgical techniques employed in different types of vascular rings, highlighting the importance of individualized surgical approaches [1]. In the type of vascular ring described (i.e., RAA with aberrant LSA and LLA), the two primary factors causing tracheoesophageal compression are the space-occupying effect of the KD and the sling-like effect of the aberrant LSA [9]. Therefore, to relieve the compression, the surgical repair should aim to divide the arterial ligament, obliterate the KD, as it can be an indication for reoperation, and transfer the LSA to the LCA [9,10]. In this case, a limited left posterior thoracotomy was the chosen surgical approach considering the absence of additional intracardiac abnormalities, allowing the visualization of the vascular ring’s anatomy. During the surgical procedure, the LLA was divided, the KD was obliterated, and despite the patient’s weight of 8.2 kg, the LSA was successfully transferred to the LCA. The re-implantation of the LSA in this type of vascular ring is feasible even for infants, having a good patency, as confirmed by the follow-up CTA at mid-term. EndoFLIP^TM^ technology, employed for assessing the esophageal cross-sectional area via high-resolution impedance planimetry during volume-controlled distension, has been utilized in the pediatric population for diagnosing and managing esophageal disorders, and it can serve for the intraoperative assessment of the esophageal compression release during vascular ring repair [11]. In our patient, endoscopy and EndoFLIP^TM^ technology provided objective intraoperative measurements of the esophageal narrowing and impedance improvement.

The surgical repair of vascular rings has a very low risk of morbidity and mortality, with favorable mid- and long-term outcomes [5,10,12]. The obliteration of the KD and reimplantation of the LSA achieve good outcomes in eliminating respiratory and esophageal symptoms after the vascular ring repair [9,10,13,14]. Since complications such as aortic dissection and aorta rupture have been reported after KD resection, it is recommended that a KD greater than 1.5 times the origin of the LSA with obvious compression of the trachea should be operated on [10,15]. After repairing the defect, patients should be followed up with regular clinic evaluations to assess symptomatic improvement and exclude postoperative complications. Imaging studies should be conducted to ensure the patency of the LSA reimplantation and the absence of tracheal deformities or esophageal narrowing. Specifically, patients should return for control visits at 1-, 3-, 6-, and 12-months post-surgery, and then annually if no complications arise. Precautions include monitoring for respiratory or swallowing difficulties and regular imaging to ensure there is no recurrence of compression or other complications. It is recommended to continue regular follow-ups throughout life, especially during periods of rapid growth.

This case underscores the significance of managing symptomatic vascular rings via early surgical repair and introduces an alternative method for the objective intraoperative assessment of esophageal compression relief in patients with vascular rings. This approach represents a safer intervention compared to the higher risks associated with the potential complications resulting from tracheoesophageal compression.

## Figures and Tables

**Figure 1 diagnostics-14-01736-f001:**
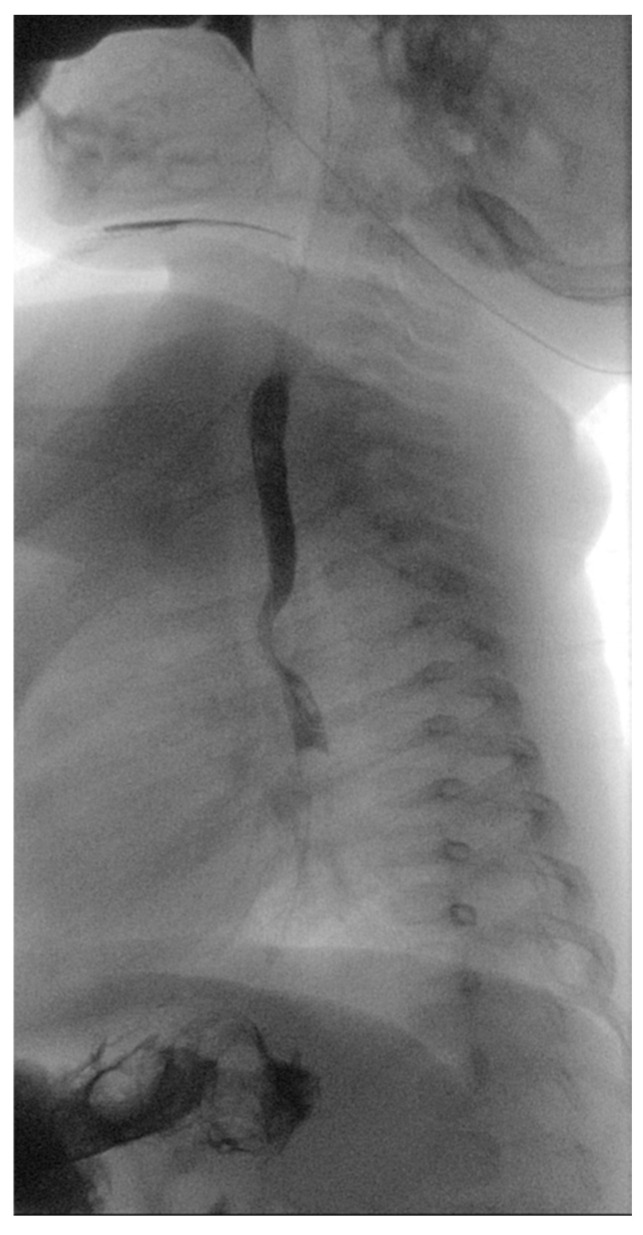
Barium swallow test. Posterior indentation of the proximal thoracic esophagus.

**Figure 2 diagnostics-14-01736-f002:**
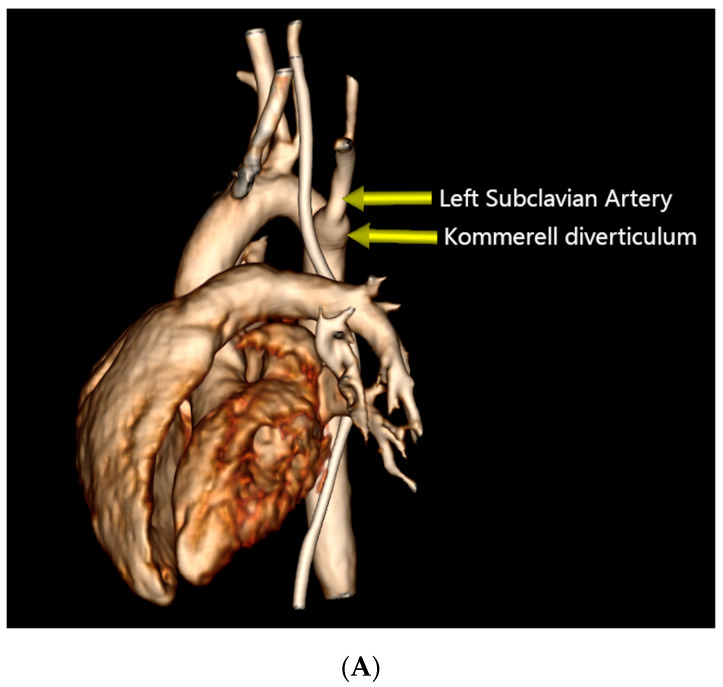
Preoperative Computed Tomography Angiography. (**A**) Three-dimensional reconstruction of an RAA with the LCA as the first branch and the retroesophageal aberrant LSA as the fourth branch, originating from a KD. (**B**) Sagittal view demonstrating esophageal narrowing and the presence of a nasogastric tube. (**C**) Coronal view showing the aneurysmal dilation of the base of the LSA.

**Figure 3 diagnostics-14-01736-f003:**
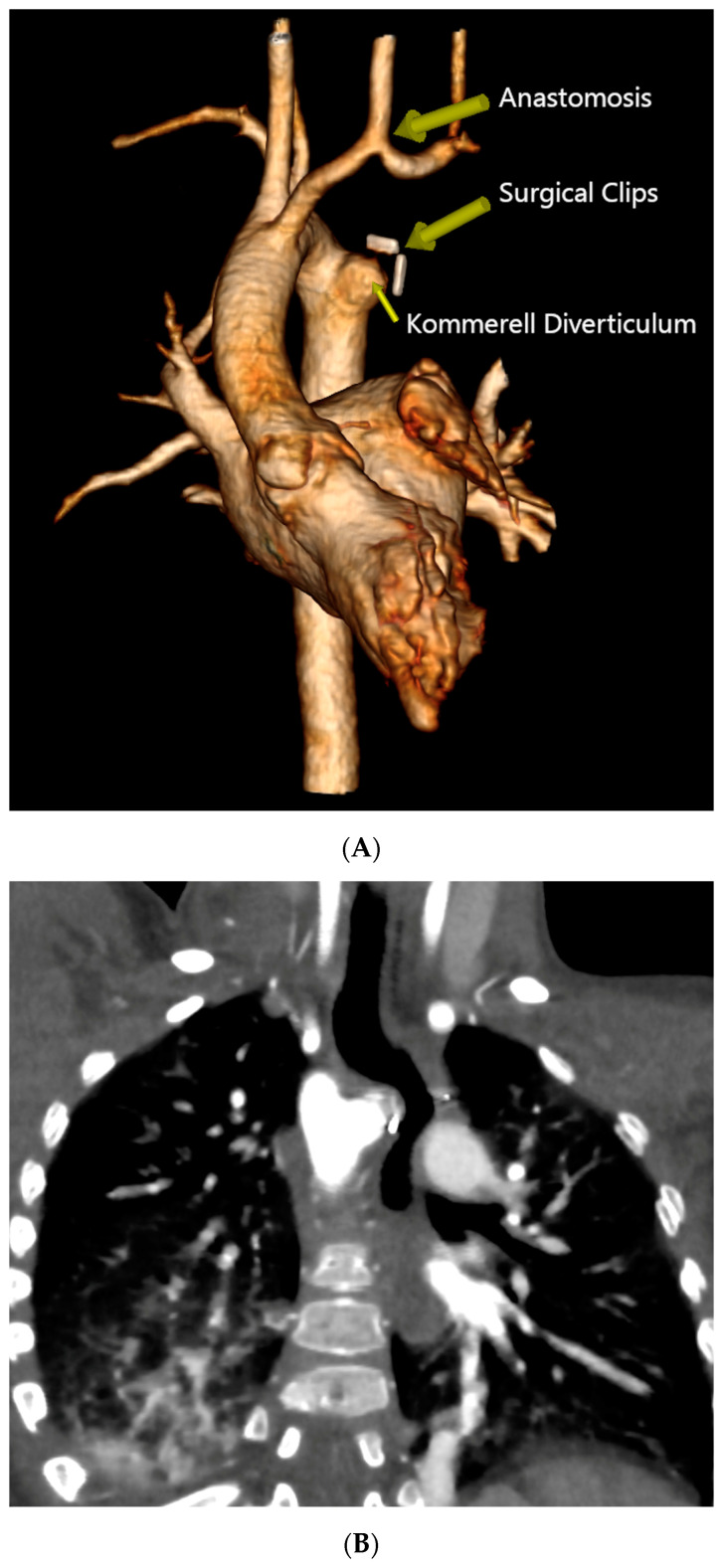
Postoperative Computed Tomography Angiography. (**A**) Three-dimensional reconstruction showing the RAA status post KD plication and LSA re-implantation into the LCA, evidencing patency of the re-implanted artery and an outpouching of the KD’s remnant with a surgical clip at the distal end. (**B**) Coronal view demonstrating esophageal displacement without evident narrowing. (**C**) Coronal view evidencing no tracheal deformities.

## Data Availability

Data are contained withing the article.

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
