# Peer review of "Vascular Ring Surgical Repair: Re-Implantation of the Left Subclavian Artery into the Left Carotid Artery in a Pediatric Patient"

_diagnostics, 2024, doi:10.3390/diagnostics14161736_

Round 1

Reviewer 1 Report

Comments and Suggestions for Authors

Dear authors,

Thank you for your interest in publishing such an interesting case. Both from an academic point of view and from a personal point of view. You provided a good example of how to handle such cases therapeutically in your case description.

I have some concerns that I would like to address to you.

I did not understand if the fetal morphology was performed and if so, how come this intrauterine defect was not observed?

What is the next course of action, after repairing the defect? How often should she return for control? What are the precautions for such a case throughout life? Do you recommend certain special measures for such a case?

I  have two suggestions for your article: 1) expand more the part of the existing literature regarding aortic arch cases and their management; and 2) increase the number of references, which of course will naturally follow point 1. Allow me to make some suggestions in this area: 10.1093/ejcts/ezab432, https://doi.org/10.1016/j.athoracsur.2019.06.076, https://doi.org/10.1016/j.jtcvs.2004.10.044, 10.7759/cureus.24623, 10.21037/tp-23-39, https://doi.org/10.1016/S1010-7940(02)00213-0, https://doi.org/10.1016/j.avsg.2021.04.036, https://doi.org/10.1177/21501351231194256,  10.4103/apc.apc_33_21. 

Congratulations on solving the case!

Comments on the Quality of English Language

Overall, it is okay. Minor editing is necessary. 

Author Response

Comment 1: I did not understand if the fetal morphology was performed and if so, how come this intrauterine defect was not observed?

Response 1: 

  • Thank you for raising this point. Vascular rings are often difficult to diagnose prenatally as they may not always be evident in routine fetal morphology scans, particularly if they do not cause significant compression of the trachea or esophagus at that stage. Prenatal diagnosis is usually performed with fetal echocardiography, and while some congenital heart defects can be detected, vascular rings may not always be identified, and the rate of missed diagnosis and misdiagnosis is still very high. This could be due to the fetal position, the size and visibility of the vascular ring, or the absence of symptoms until after birth, which was the case here. These points were included in the manuscript and highlighted for the reviewer’s recognition.

Comment 2: What is the next course of action, after repairing the defect? How often should she return for control? What are the precautions for such a case throughout life? Do you recommend certain special measures for such a case?

Response 2: 

  • Thank you for asking this question. After repairing the defect, the patient was followed up with regular clinic evaluations to assess symptomatic improvement and exclude postoperative complications, and with imaging studies to check for patency of the left subclavian artery reimplantation, and absence of tracheal deformities or esophageal narrowing. Specifically, the patient should return for control visits at 1-, 3-, 6-, and 12-months post-surgery, and then annually if no complications arise. Precautions include monitoring for respiratory or swallowing difficulties and regular imaging to ensure no recurrence of compression, patency of the arterial reimplantation, or the presence of any complication. It is recommended to continue regular follow-ups throughout life, especially during periods of rapid growth. These points were included in the manuscript and highlighted for the reviewer’s recognition.

Comment 3: I  have two suggestions for your article: 1) expand more the part of the existing literature regarding aortic arch cases and their management; and 2) increase the number of references, which of course will naturally follow point 1. Allow me to make some suggestions in this area: 10.1093/ejcts/ezab432, https://doi.org/10.1016/j.athoracsur.2019.06.076, https://doi.org/10.1016/j.jtcvs.2004.10.044, 10.7759/cureus.24623, 10.21037/tp-23-39, https://doi.org/10.1016/S1010-7940(02)00213-0, https://doi.org/10.1016/j.avsg.2021.04.036, https://doi.org/10.1177/21501351231194256,  10.4103/apc.apc_33_21. 

Response 3: 

  • Thank you for this comment and for the provided literature. The manuscript was expanded mainly in the discussion section, covering vascular ring literature, aortic arch cases (including double aortic arch and right aortic arch with aberrant left subclavian artery and left ligamentum arteriosum), diagnostic evaluation both prenatally and postnatally, management, and the follow-up process with its precautions and recommendations.
  • The following points were included in the manuscript:
    • Vascular rings are congenital aortic arch anomalies that result in compression of the trachea, esophagus, or both [2]. The most reported types of vascular ring are the double aortic arch (DAA) and the RAA with aberrant LSA and LLA, the latter being usually associated with a Kommerell’s diverticulum [4]. Patients presenting with a DAA commonly exhibit symptoms in the first year of life due to severe respiratory compromise secondary to significant tracheal compression [4]. However, patients with an RAA with an aberrant LSA and LLA typically start having symptoms within the first few years of life, usually presenting as a combination of respiratory and esophageal manifestations [4]. Hence, symptoms presentation depends on the type of vascular ring, the degree of compression, and the patient’s age, ranging from respiratory distress in newborns to feeding difficulties in older children [2, 5].
    • Traditionally, the initial evaluation of vascular rings involves chest X-ray, barium swallow, and echocardiography [1-4]. However, cross-sectional imaging techniques like Multidetector Computed Tomography (MDCT) and Magnetic Resonance Imaging (MRI) are necessary for detailed anatomical assessment. MDCT offers excellent visualization of vascular anatomy and is often preferred over MRI due to its speed and clarity without requiring sedation [1]. Although prenatal diagnosis of vascular rings has improved with fetal echocardiography and advanced 3D sonography, which provide clearer views of the aortic arch position relative to the trachea, the rate of missed diagnosis and misdiagnosis is still very high [1, 8]. A recent study by Zhou et al. (2023) demonstrated that dynamic sequential cross-sectional observation (SCS) can accurately diagnose congenital vascular rings in the prenatal period, although it supports expectant management until birth [4, 7].
    • Yoshimura et al. (2020) discuss the various surgical techniques employed in different types of vascular rings, highlighting the importance of individualized surgical approaches [1].
    • Surgical repair of vascular rings has a very low risk of morbidity and mortality, with favorable mid- and long-term outcomes [5, 10, 12]. Obliteration of the KD and reimplantation of the LSA achieve good outcomes in eliminating respiratory and esophageal symptoms after the vascular ring repair [9, 10, 13, 14]. Since complications such as aortic dissection and aorta rupture have been reported after KD resection, it is recommended that a KD greater than 1.5 times the origin of the LSA with obvious compression of the trachea should be operated [10, 15]. After repairing the defect, patients should be followed up with regular clinic evaluations to assess symptomatic improvement and exclude postoperative complications. Imaging studies should be conducted to ensure the patency of the LSA reimplantation, absence of tracheal deformities, and no esophageal narrowing. Specifically, patients should return for control visits at 1-, 3-, 6-, and 12-months post-surgery, and then annually if no complications arise. Precautions include monitoring for respiratory or swallowing difficulties and regular imaging to ensure no recurrence of compression or other complications. It is recommended to continue regular follow-ups throughout life, especially during periods of rapid growth.
  • The following references were included:
    • Fisenne DT, Burns J, Dhar A. Feeding Difficulties Following Vascular Ring Repair: A Contemporary Narrative Review. Cureus. 2022 Apr 30;14(4):e24623. doi: 10.7759/cureus.24623. eCollection 2022 Apr.
    • Binsalamah ZM, Ibarra C, John R, et al. Contemporary Midterm Outcomes in Pediatric Patients Undergoing Vascular Ring Repair. Ann Thorac Surg. 2020 Feb;109(2):566-572. doi: 10.1016/j.athoracsur.2019.06.076. Epub 2019 Aug 14.
    • Zhou Y, Zhou Y, Yu T, et al. Vascular ring: prenatal diagnosis and prognostic management based on sequential cross-sectional scanning by ultrasound. BMC Pregnancy Childbirth. 2023 May 2;23(1):308.doi: 10.1186/s12884-023-05637-y.
    • Yu D, Guo Z, You X, et al. Long-term outcomes in children undergoing vascular ring division: a multi-institution experience. Eur J Cardiothorac Surg. 2022 Feb 18;61(3):605-613.doi: 10.1093/ejcts/ezab432.
    • Charbonneau P, Fabre D, Le Bret E, et al. A Ten-year Single-center Surgical Experience With Symptomatic Complete Vascular Rings. Ann Vasc Surg. 2022 Jan:78:70-76.doi: 10.1016/j.avsg.2021.04.036. Epub 2021 Jun 25.
    • Cheong D, Jhaveri S, Meyer DB, et al. Association and Repair of Right Aortic Arch With Aberrant Left Subclavian Artery With Subclavian Stenosis. World J Pediatr Congenit Heart Surg. 2024 Jan;15(1):133-136.doi: 10.1177/21501351231194256. Epub 2023 Sep 20.
    • Backeer CL, Mavroudis C, Rigsby CK, et al. Trends in vascular ring surgery. J Thorac Cardiovasc Surg. 2005 Jun;129(6):1339-47.doi: 10.1016/j.jtcvs.2004.10.044.
    • Tanaka A, Milner R, Ota T. Kommerell’s diverticulum in the current era: a comprehensive review. Gen Thorac Cardiovasc Surg. 2015 May;63(5):245-59.doi: 10.1007/s11748-015-0521-3. Epub 2015 Jan 31.

Reviewer 2 Report

Comments and Suggestions for Authors

The subject is interesting. The quality of English language is good. However the manuscript has to be improved. Structure is incomplete: it may be organized as usual with Introduction, Clinical presentation, Discussion and Conclusion. Iconography must be improved; a 3d reconstruction of preoperative CT scan is suggested and intraoperative photos are recommended.

Comments on the Quality of English Language

The subject is interesting. The quality of English language is good. However the manuscript has to be improved. Structure is incomplete: it may be organized as usual with Introduction, Clinical presentation, Discussion and Conclusion. Iconography must be improved; a 3d reconstruction of preoperative CT scan is suggested and intraoperative photos are recommended.

Author Response

Comment 1: Structure is incomplete: it may be organized as usual with Introduction, Clinical presentation, Discussion and Conclusion.

Response 1: 

  • Thank you for raising that point. The manuscript was originally submitted using the mentioned structure. However, one of the journal’s editors suggested removing the section titles. The manuscript has now been updated to include those titles for more clarity within the different sections.

Comment 2: Iconography must be improved; a 3d reconstruction of preoperative CT scan is suggested and intraoperative photos are recommended.

Response 2:

  • Thank you for this concern. Both a 3D reconstruction of the preoperative and postoperative CT scans is available in the manuscript as figures 2 and 3, and in the supplementary material as Videos S1 and S2. A clarification in the figure legends was made to identify the preoperative and postoperative imaging. Thank you for the suggestion of intraoperative photos. Unfortunately, they are not available at this time.
    • Figure 2. Preoperative Computed Tomography Angiography. (A) 3D reconstruction of a RAA with the LCA as the first branch, and the retroesophageal aberrant LSA as the fourth branch originating from a KD. (B) sagittal view demonstrating esophageal narrowing and the presence of a nasogastric tube. (C) coronal view showing the aneurysmal dilation of the base of the LSA.
    • Figure 3. Postoperative Computed Tomography Angiography. (A) 3D reconstruction showing a RAA status post KD plication and LSA re-implantation into the LCA, evidencing patency of the re-implanted artery and an outpouching of the KD’s remnant with a surgical clip in the distal end. (B) coronal view demonstrating esophageal displacement without evident narrowing. (C) coronal view evidencing no tracheal deformities.
    • Video S1. Preoperative Computed Tomography Angiography, 3D reconstruction.
    • Video S2. Postoperative Computed Tomography Angiography, 3D reconstruction.

Round 2

Reviewer 2 Report

Comments and Suggestions for Authors

The main question addressed by the Authors is the surgical repair of vascular rings with intraoperative endoscopic assessment of improvement in esophageal narrowing. Preoperative evaluation and follow-up methods are also described. Due to the rarity of the condition, case reports are encouraged, and this one describes surgical technique and management well offering o good alternative to compare with. No further controls should be considered. The document answers all questions posed by the previous review.  The structure aìof the manuscript has been improved. References appear appropriate. Figures have been improved according to previous suggestions. 

Line 32: “by encircling and compressing” reads better.

The manuscript seems ready for publication to me.

Congrats to Authors.

Comments on the Quality of English Language

The English language is good. 

Line 32: “by encircling and compressing” reads better.